# Moss enables high sensitivity single-nucleotide variant calling from multiple bulk DNA tumor samples

Chuanyi Zhang ⓘ [1], Mohammed El-Kebir ⓘ [2✉] & Idoia Ochoa ⓘ [1,3✉]

Intra-tumor heterogeneity renders the identification of somatic single-nucleotide variants (SNVs) a challenging problem. In particular, low-frequency SNVs are hard to distinguish from sequencing artifacts. While the increasing availability of multi-sample tumor DNA sequencing data holds the potential for more accurate variant calling, there is a lack of high-sensitivity multi-sample SNV callers that utilize these data. Here we report Moss, a method to identify low-frequency SNVs that recur in multiple sequencing samples from the same tumor. Moss provides any existing single-sample SNV caller the ability to support multiple samples with little additional time overhead. We demonstrate that Moss improves recall while maintaining high precision in a simulated dataset. On multi-sample hepatocellular carcinoma, acute myeloid leukemia and colorectal cancer datasets, Moss identifies new low-frequency variants that meet manual review criteria and are consistent with the tumor's mutational signature profile. In addition, Moss detects the presence of variants in more samples of the same tumor than reported by the single-sample caller. Moss' improved sensitivity in SNV calling will enable more detailed downstream analyses in cancer genomics.

[1] Department of Electrical and Computer Engineering, University of Illinois at Urbana-Champaign, Urbana, IL, USA. [2] Department of Computer Science, University of Illinois at Urbana-Champaign, Urbana, IL, USA. [3] Department of Electrical Engineering, University of Navarra, Tecnun, San Sebastian, Spain. ✉email: melkebir@illinois.edu; idoia@illinois.edu

Cancer results from an evolutionary process where somatic mutations accumulate in a population of cells[1]. While germline mutations are inherited, somatic mutations occur during the lifetime of an individual. Somatic mutations vary in genomic scale, ranging from single-nucleotide variants (SNVs) that affect individual bases, structural variants (SVs), and copy-number aberrations (CNAs) that affect large genomic regions such as chromosome arms, to whole-genome duplications that affect the entire genome. Importantly, groups of tumor cells, or clones, vary in their complement of somatic mutations—a phenomenon known as intra-tumor heterogeneity. To understand mechanisms of tumorigenesis and devise personalized treatment plans, it is important to fully characterize the extent of intra-tumor heterogeneity of a cancer. This begins with accurately calling the set of somatic mutations that are present in a tumor, which is the first step in many important downstream analyses in cancer genomics, including identifying mutations that drive cancer progression[2], reconstructing the evolutionary history of a tumor[3,4], predicting the response to immunotherapy[5], identifying (exposures to) mutational signatures[6,7] and reconstructing repeated patterns of cancer evolution[8] and metastasis[9]. Inaccurate or incomplete variant calling may lead to incorrect conclusions in downstream cancer genomics analyses.

The key challenge in variant calling arises due to limitations in current sequencing technologies, which are unable to sequence complete genomes from end to end. These technologies are applied to bulk sequencing samples and yield DNA reads that are orders of magnitudes shorter than the genome in question. To overcome this challenge, current variant callers take as input a mapping of the input sequence reads to a reference genome, from which they identify variants while accounting for sequencing and mapping errors. In germline variant calling, the used reference genome is the reference genome of the species in question. There are two additional challenges in the somatic variant calling of tumor samples. First, the goal is to identify variants that do not occur in the germline and are unique to the tumor. To accomplish this, a matched normal sample is sequenced in addition to one or more samples from the tumor. Somatic variant callers use the matched normal sample to identify germline variants and obtain a new reference genome, which is used in turn to identify somatic variants. Second, the presence of intra-tumor heterogeneity results in somatic mutations with varying variant allele frequencies (VAFs) in the tumor samples. While germline mutations typically have a small number of frequencies depending on their zygosity (e.g., a frequency of 0, 0.5, and 1 for diploid organisms), somatic mutations may have VAFs that span the full range of frequencies between 0 and 1. Somatic variant callers must take this heterogeneity into account in addition to the

presence of sequencing and mapping errors. It is particularly challenging to distinguish low-frequency somatic variants from sequencing and mapping artifacts—this is especially the case for SNVs.

Several methods have been proposed for somatic single-nucleotide variant calling. Mutect2[10] from the Genome Analysis Tool Kit (GATK) firstly performs local assembly and read-haplotype alignment, then approximates the likelihood function of a genotype with a Bayesian model. Strelka2[11], developed by Illumina, models allele frequencies for the normal and tumor samples as latent variables and computes posterior probabilities by marginalizing over the frequencies. MuSE[12] uses a Markov substitution model to estimate the equilibrium frequencies of alleles and then computes a cutoff from a sample-specific error model. CaVEMan[13] takes as input the aligned reads, copy numbers, and contamination in the normal sample by the tumor, and then uses expectation-maximization to calculate the probability of an SNV. Other SNV callers include Lancet[14], Platypus[15], LoFreq[16], and many others. Importantly, the aforementioned variant callers take as input a single tumor sample only. However, multi-sample datasets[17] enable a more precise characterization of the clones present in a tumor as well as the tumor's evolutionary history[18,19]. This, coupled with decreasing sequencing costs and the availability of new profiling techniques, such as liquid biopsies, have led to an increasing availability of multi-sample data. Current single-sample SNV callers are unable to unlock the potential of these data, which enable more accurate variant calling because the probability of the same sequencing error occurring in all tumor samples at the same locus decreases significantly with an increasing number of tumor samples (Fig. 1a).

Here we propose Moss, a somatic SNV caller that leverages the additional information afforded by multi-sample tumor data to enable improved sensitivity SNV calling compared to existing single-sample somatic SNV callers. Moss is designed to be a light-weight versatile tool that turns any single-sample caller of choice into a multi-sample caller (Fig. 1b). Moss uses a Bayesian model that accounts for multiple tumor samples to identify SNVs from a candidate set generated by running an existing single-sample SNV caller under lenient conditions on each tumor sample in isolation. While the majority of somatic SNV callers support a single tumor sample only, there are two exceptions: the most recent version of Mutect2 (GATK version 4.1) and multisnv[20]. Using simulated data we demonstrate that Moss outperforms these two multi-sample callers as well as improves upon widely used single-sample callers. Specifically, our simulations show that running Moss in conjunction with Mutect2 (single-sample mode) or Strelka2 accurately recovers variants with low VAF, thereby increasing recall while maintaining high precision. On two multi-

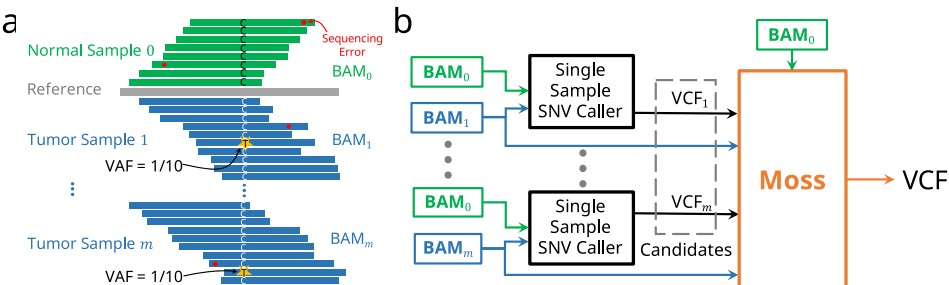

**Fig. 1 Moss extends current single-nucleotide variant callers by leveraging multiple samples to improve recall. a** When multiple tumor samples are analyzed simultaneously, single-nucleotide variants (SNVs, yellow stars) with low variant allele frequency (VAF) can be more easily distinguished from sequencing errors (red dots) than when tumor samples are analyzed in isolation. **b** Workflow overview of Moss, taking as input BAM (or SAM) files of a matched normal sample (subscript 0) and *m* tumor samples, along with *m* VCF (variant call format) files obtained by any existing single-sample SNV caller. The output of Moss is an aggregate VCF file containing the called SNVs present in the *m* tumor samples.

sample tumor datasets, we find that the majority of the additional low-frequency SNVs identified by Moss are well supported by cancer-type specific mutational signatures, as well as pass manual review criteria[21]. Similarly, we improve recall in an acute myeloid leukemia benchmark dataset with a gold list identified by high-coverage targeted sequencing[22]. Finally, we find that Moss adds little overhead in terms of running time compared to single-sample callers. Moss' improved sensitivity in calling low-frequency SNVs will enable more detailed downstream analyses in cancer genomics.

## Results

**Overview of the method**. We consider $m$ tumor samples and a matched normal sample (Fig. 1a). In our proposed workflow, a single-sample SNV caller is run independently on each of the $m$ tumor/normal pairs to obtain a permissive candidate call set (Fig. 1b). Moss then extracts the set of candidate loci by taking the union of the positions of all the SNV records in the VCF (variant call format) files (obtained under filtering criteria that are more permissive than the default parameters including those that do not pass the set filters), as well as the normal alleles inferred by the base caller. This information together with the original and realigned BAM (or SAM) files of the $m$ tumor samples and the matched normal sample form the input to Moss.

Moss evaluates each candidate locus independently, and computes the somatic SNV probability, i.e., the posterior probability of whether a locus contains an SNV, and the corresponding tumor allele if an SNV is present. In line with Strelka2[11], we assume the presence of at most one tumor allele per locus. Moss incorporates a Bayesian model to calculate the somatic SNV probability, considering the normal allele, aligned bases, and quality scores as observations and the tumor allele as the latent variable. The output of Moss is an aggregate VCF file containing the called SNVs present in the $m$ tumor samples.

To increase confidence in calls, Moss may optionally discard reads that do not meet minimum quality requirements. Then, as a final (optional) step, Moss applies several empirical filters to the newly identified variants to further reduce the false positive rate (Methods). For example, variants with a strand bias that is likely caused by a systemic error during sequencing are filtered out. To speed up the computation and to make sure the posterior probability does not vanish with an increased number of samples, Moss excludes samples whose reads are identical to the normal allele at the considered locus in the computation of the posterior probability. In addition, if only one tumor sample contains reads with the variant allele, we defer to the single-sample caller and call the SNV only if it was called by the original caller.

Moss is implemented in C++ and utilizes the HTSlib library[23] for accessing SAM/BAM and VCF files. Moss is also equipped with Python scripts providing an easy configuration and running interface. The code and scripts are available at https://github.com/elkebir-group/Moss. See "Methods" for further detail.

**Moss improves accuracy in simulated data**. We first evaluate the accuracy of Moss on a synthetic dataset, showing that Moss increases recall without loss of precision of two widely used single-sample SNV callers Strelka2[11] and Mutect2[10]. We simulate Illumina sequencing of $m = 5$ bulk DNA samples of chromosome 20 as follows (Fig. 2a). First, we generate a matched normal sample by adding germline SNPs from dbSNP[24] to chromosome 20 of the human reference genome GRCh38 p12. Next, we randomly generate somatic mutations with a probability of 0.001. Then, we split the mutations into four groups at random, forming a simple linear phylogeny tree with four clones. After inserting 75,958 mutations in total into clones with MASCoTE[25], we

generate $m = 5$ samples with different mixture ratios and a mean coverage of either 30× or 60×.

We run Strelka2 and Mutect2 (with default settings) independently on each of the 5 tumor/normal pairs. Additionally, to assess the performance of the single-sample callers on multi-sample data, we take the union of the SNVs that were called in the individual samples. We then run each single-sample SNV caller under permissive conditions to obtain the input candidate sets for Moss. Specifically, Moss takes the resulting VCF files and the realigned BAM files as input. We plot the precision-recall (PR) curves obtained by each method (Fig. 2b). Here, recall refers to the fraction of simulated SNVs that were called and precision is the fraction of calls that correspond to simulated SNVs. Specifically, we generate the PR curves by adjusting a method-specific threshold for a given feature in the VCF file, i.e., "feature EVS" for Strelka2, "TLOD" for Mutect2 and the "somatic probability" for Moss.

We find that, compared to Mutect2 and Strelka2 run in isolation, Moss improves recall without loss of precision (Fig. 2b). This is also evidenced by the $F_1$ score, which is the harmonic mean of recall and precision. For instance, we see that the combination of Strelka2 and Moss achieves a maximum $F_1$ score of 0.93 compared to a maximum $F_1$ score of 0.89 for Strelka2 run in isolation (Fig. 2b). We obtain similar results using $m \in \{2, 3, 4\}$ samples and coverage of 30× (Supplementary Fig. 1). Moreover, the run time of Moss is negligible compared to the single-sample callers (Supplementary Fig. 2). Finally, we compare the performance of Moss to a recently released version of Mutect2 with multi-sample capabilities. While this newer version of Mutect2 has similar performance in terms of recall and precision to the combination of single-sample Mutect2 and Moss (Supplementary Fig. 1), we find that multi-sample Mutect2 takes much longer to run (Supplementary Fig. 2). By contrast, multisnv[20] achieves worse recall than Moss (Supplementary Fig. 1) and has the longest running time of all methods (Supplementary Fig. 2). In summary, our simulations show that Moss improves recall without loss of precision and has negligible run time overhead compared to the original single-sample SNV callers.

**Evaluating Moss in a hepatocellular carcinoma dataset**. We further evaluate the performance of Moss on a hepatocellular carcinoma (HCC) tumor[26], for which 23 tumor biopsies, as well as a tumor-adjacent, matched normal sample were sequenced at an average depth of 74.4× (Fig. 3a). Due to the large number of samples, we were not able to run multi-sample Mutect2 and multisnv on this dataset. Hence, in the following, we focus on the performance of Moss when run in conjunction with single-sample Mutect2 and Strelka2. Specifically, we run Strelka2 and Mutect2 in isolation and in conjunction with Moss. The comparison of Moss to the single-sample callers takes the union of the called SNVs under the caller's default criteria. As before, the input provided to Moss consists of the union of all candidate SNVs identified by the single-sample caller under permissive conditions as well as aligned reads from normal and tumor samples. Due to space constraints, we only report results for Mutect2 in the main text and refer to Supplementary Fig. 3 for results using Strelka2.

Similarly to the simulations, we find that Moss identifies additional SNVs when applied after Mutect2 (Fig. 3b) and Strelka2 (Supplementary Fig. 3a). In particular, Moss calls 466 additional SNVs when run in conjunction with Mutect2, retaining all but one variant identified by single-sample Mutect2. For the lost variant, we observe that the base quality of the mutated bases is significantly lower than that of the non-mutated bases, indicating a potential sequencing and/or mapping artifact

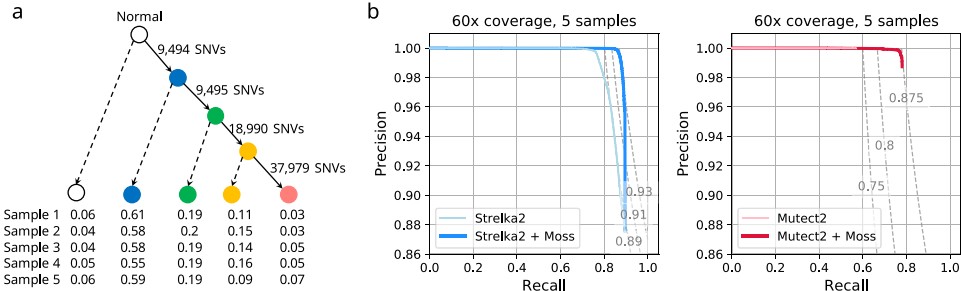

**Fig. 2 Moss improves recall without loss of precision on synthetic data. a** Linear phylogenetic tree with the root representing the normal clone and four additional tumor clones. Edge label indicates number of SNVs (single-nucleotide variants) newly introduced into each tumor clone. The table shows the prevalence of each clone in each sample. **b** Precision-recall curves of the union of SNVs identified by Strelka2 (light blue) and Mutect2 (light red) when applied independently to samples of a simulated bulk DNA sequencing dataset with 5 samples and 60× coverage, as well as when Moss is applied in conjunction with these methods (blue and red). The dashed lines represent $F_1$ score isolines (i.e., harmonic mean between recall and precision).

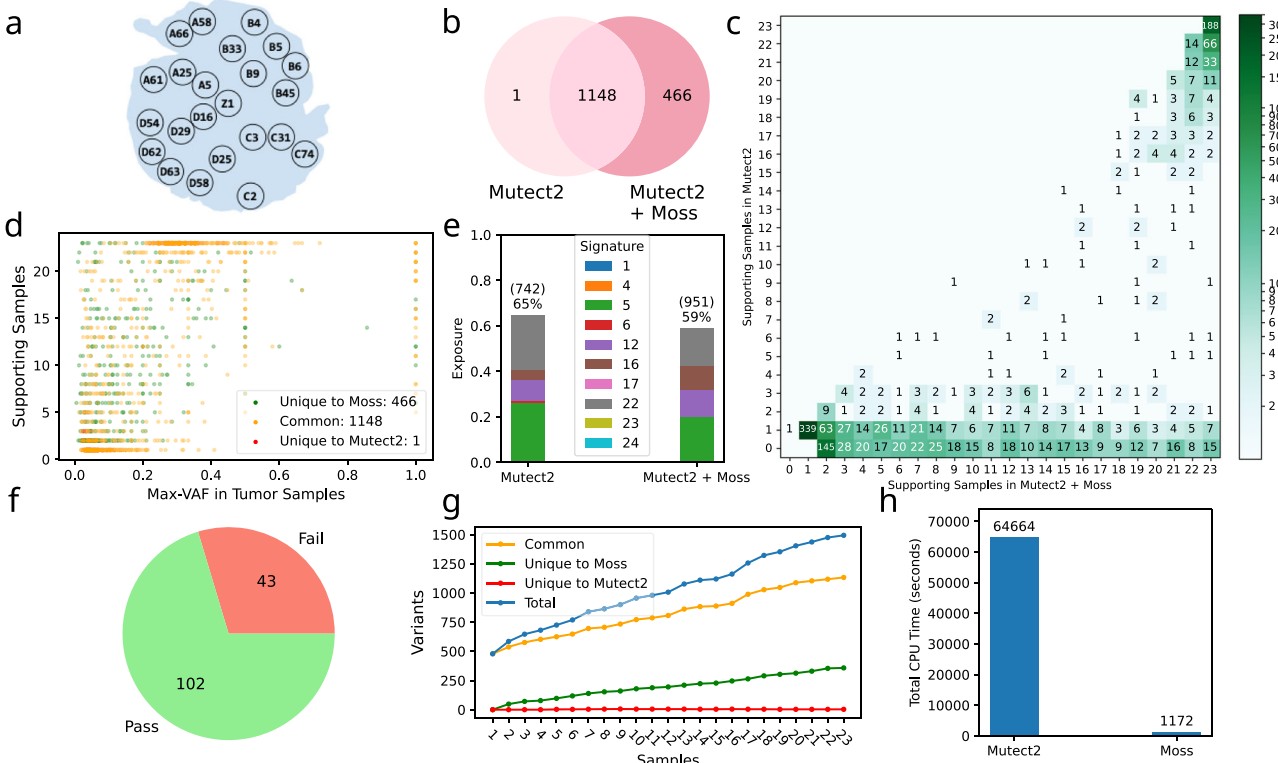

**Fig. 3 Moss recovers high-quality somatic variants missed by a single-sample variant caller in a hepatocellular carcinoma (HCC) dataset[26]. a** Ling et al.[26] performed whole-exome sequencing (WES) for 23 biopsies of an HCC tumor. **b** Venn diagram comparing the call set of Mutect2 when run in isolation and when run in conjunction with Moss. Moss identifies 466 new variants while retaining all variants (but one) identified by Mutect2. **c** The number of supporting samples identified by Moss (x-axis) and the single-sample caller Mutect2 (y-axis) for each variant, showing that Moss increases the number of supporting samples for 36% of variants (586 variants, spatial distribution provided in Supplementary Fig. 5). Variants uniquely recovered by Moss correspond to entries with y-axis equal to 0. **d** The number of samples identified by Moss to contain a variant as a function of the variant's largest frequency across all tumor samples, showing that most of the variants recovered by Moss have low VAF. Color indicates whether the variant is common to Mutect2 and Moss (yellow), or unique to Moss (green) or Mutect2 (red). **e** Exposure to mutational signatures (indicated by color) of liver tumor for different methods. Applying Moss increases the number of variants explained by the mutational signatures (951 vs 742). **f** Out of the 145 SNVs newly called by Moss in exactly $m = 2$ samples, 102 SNVs passed the manual review (Supplementary Fig. 7). **g** Analyzing multiple samples simultaneously increases the number of recovered variants significantly. **h** Moss adds almost no overhead in run time as compared to Mutect2. Similar results are observed for Strelka2 (Supplementary Fig. 3).

(Supplementary Fig. 4). After calling a variant, Moss identifies which samples support it by evaluating the corresponding BAM files. Moss, therefore, identifies all samples supporting a given variant, while Mutect2 generally does not (Fig. 3c, with the

majority of entries below the diagonal line). In particular, Moss increases the number of supporting samples for 586 (36%) of variants identified by single-sample Mutect2, reducing the number of SNVs that are unique to specific spatial locations of

the tumor (Supplementary Fig. 5). We observe similar behavior with Strelka2 (Supplementary Fig. 3b).

We further find that the variants identified uniquely by Moss typically have low VAF. In particular, most of these variants have a VAF no greater than 0.3 across the supporting tumor samples (Fig. 3d). We observe similar results when Moss is run in conjunction with Strelka2 (Supplementary Fig. 3d). These findings support our claim that jointly analyzing multiple tumor samples can help in recovering low-VAF variants. To ensure the absence of germline leakage, we verify that the recovered variants have low-VAF in the normal sample both for Moss with Strelka2 and Mutect2 (Supplementary Fig. 6). For example, the set of called variants recovered by Moss when run in conjunction with Mutect2 do not exhibit a VAF in the normal sample greater than 0.06. This suggests that the variants recovered by Moss are not germline mutations.

While the simulations had ground truth, there is no ground truth set of SNVs available for the hepatocellular carcinoma (HCC) dataset. Hence, to partially verify that the set of variants identified by Moss is accurate, we analyze their mutational patterns. In particular, we calculate exposures to the 30 COSMIC v2 mutational signatures of the mutations identified by Moss as well as the single-sample callers[27–30]. We are interested in the exposure percentages of liver cancer signatures. A higher or identical percentage of liver cancer signature exposures in the SNVs identified by Moss compared to those identified by the single-sample caller indicates that the novel SNVs have the same etiology as the original set of SNVs, which is indicative of their validity. We perform this analysis using deconstructSig[31], normalizing the counts of the 96 types of mutations. The call set of Moss when run in conjunction with Mutect2 contains Signatures 5, 12, 16, and 22, which have been determined to occur in liver cancer[28] (Fig. 3e). Although the total exposure is lower than when Mutect2 is run in isolation (59% vs 65%), more variants are explained by liver cancer-related signatures (about 951 SNVs for Moss and 742 for Mutect2). Higher total exposure is obtained when Strelka2 is run in conjunction with Moss rather than in isolation (54.2 vs 52.3%) (Supplementary Fig. 3c).

For further validation, we perform manual review of the variants called by Moss in exactly two samples but not called by Mutect2 in any sample. We follow the procedure of Barnell et al.[21]. Prior to applying Moss' filtering criteria, there are 166 such variants (Supplementary Fig. 7a). After filtering (described in "Methods"), Moss identifies 145 variants. Manual review suggests that 102 of these are true SNVs (Fig. 3f, Supplementary Fig. 7b, and Supplementary Data 1). Of the remaining $145 - 102 = 43$ SNVs called by Moss, 32 were

identified as ambiguous in at least one sample. As for the 166 $- 145 = 21$ variants filtered out by Moss, 1 SNV is flagged as tumor-in-normal (TIN), 8 SNVs as empty-strand, 10 as low-tumor-support, and 7 as cluster (note that a SNV can have more than one flag). Manual review of these variants results in assigned tags that match Moss' filtering criteria (Supplementary Data 1). When analyzing all variants, the TIN filter flagged in total 183 SNVs, empty-strand 445 SNVs, low-tumor-support 772, and cluster 701 SNVs. These findings demonstrate that the implemented filters are capable of removing artifacts, and that the majority of the analyzed SNVs newly called by Moss pass manual inspection.

To showcase the benefit of analyzing multiple samples, even when only a small number of them are available, we run the single-sample callers Mutect2 and Strelka2 in isolation and in conjunction with Moss on the 23 samples incrementally. For the single-sample callers, as before, we take as the call set the union across all the (independently) analyzed samples. We observe that the number of called variants increases with the number of samples, yielding a higher recall rate (Fig. 3f and Supplementary Fig. 3e). Even with 2 or 3 samples, the number of recovered variants by Moss increases as compared to running the single-callers in each sample independently. In particular, Moss recovers 48 and 71 variants missed by Mutect2 with 2 and 3 samples, respectively, which corresponds to an increase of 8.9% and 12.3%. In order to verify Moss' ability to work on datasets with low coverage, we downsample the HCC dataset (original coverage 75×) to 30×, 20×, and 10×, and then rerun Mutect2 in isolation and in conjunction with Moss incrementally on the 23 samples. We find that Moss retains the ability to recover additional variants on low-coverage datasets (Supplementary Fig. 8).

Mutect2 requires about 18 hours to complete SNV calling in the 23 samples of the HCC dataset, whereas Moss needs less than an additional 20 min (Fig. 3g). Hence, Moss increases the run time by 1.8%. Moreover, Moss only requires 361 MB of memory whereas Mutect2 uses 2,216 MB. A similar overhead is observed when Moss is run with Strelka2 (Supplementary Fig. 9). All experiments were run on a machine with two 64 bit x86 Intel Xeon 2.20 GHz CPUs and 512 GB of memory.

**Evaluating Moss on an acute myeloid leukemia dataset with a manually curated list of SNVs.** We test the performance of Moss on an acute myeloid leukemia (AML) dataset[22]. In this dataset, a normal sample, a primary tumor sample and a relapse sample were sequenced with multiple sequencing strategies, including whole-genome sequencing (WGS, median coverage of 312×),

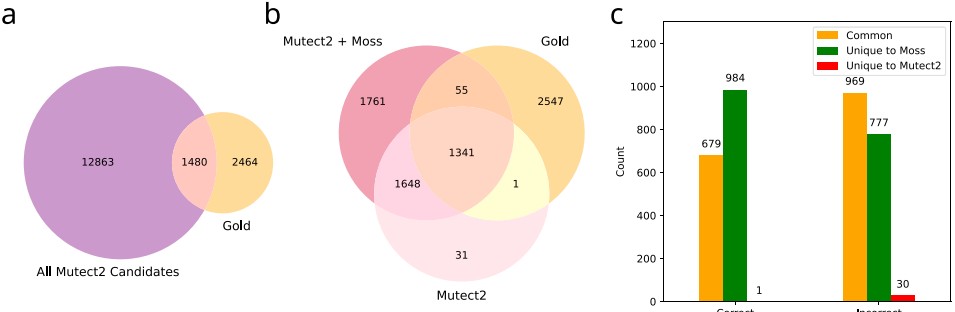

**Fig. 4 Moss increases the recall rate in an acute myeloid leukemia (AML) dataset with a manually curated "gold list" of SNVs (single-nucleotide variants)[22]. a** Venn diagram comparing the candidate set of Mutect2 with the gold list, when run independently in the whole-genome sequencing (WGS) primary tumor and relapse samples. **b** Venn diagram comparing the SNVs called by Moss and Mutect2 with the gold list. **c** Number of SNVs discovered by Mutect2 and Moss designated as "correct" or "incorrect" based on examining the VAFs in the custom targeted capture data. Color indicates whether the variant is common to Mutect2 and Moss (yellow), or unique to Moss (green) or Mutect2 (red).

whole-exome sequencing (WXS, median coverage of 433×), and custom targeted capture (median coverage of 1500×). Griffith et al. analyzed the target capture data to produce a manually curated set of high-quality SNVs designated as the gold list. Here, we run Moss in conjunction with Mutect2 on the WGS data. To enable validation, we restrict our attention to candidate SNVs occurring in the genomic regions covered by the targeted capture data (with a minimum coverage of 100× in each of the primary, relapse, and normal sample). Running Mutect2 on each WGS tumor sample in isolation yields a total of 14,343 candidates, 1480 of which occur in the gold list (Fig. 4a). While Mutect2 recalls 1342 variants from the gold list, Moss recalls 1396 variants (Fig. 4b). The single SNV from the gold list identified by Mutect2 but not by Moss has low base quality scores of the mutated base, and is subsequently filtered out (Supplementary Fig. 10). Moss additionally identifies 3409 variants not present in the gold list. To verify these SNVs, we examine their VAFs in the custom targeted capture data. Specifically, we designate a candidate SNV as "correct" if it has a VAF smaller than 0.05 in the normal sample and at least 5 reads with the variant allele in either the primary or relapse sample, otherwise the SNV is designated as "incorrect". 1663 out of the 3409 SNVs are correct, with a subset of 984 uniquely identified by Moss (Fig. 4c). Moreover, 30 out of the 31 uniquely identified by Mutect2 (that are not in the gold list) are designated as "incorrect". Hence, these findings further confirm Moss' ability to leverage multi-sample data to identify low-frequency SNVs that lead to improved sensitivity. In addition, we find that Moss decreases the number of SNVs that are unique to one of the (temporal) samples (Supplementary Fig. 11).

**Evaluating Moss in a colorectal cancer dataset**. Finally, we assess the performance of Moss run in conjunction with Strelka2[11] on a colorectal cancer (CRC) dataset[32]. This dataset includes data of two patients (denoted as Patient 43 and Patient 45), and for each of them three tumor samples and a matched normal sample (extracted from blood) were sequenced. For Patient 43, Moss recovers an additional 49,497 SNVs, while losing 318 variants called by Strelka2 (Fig. 5a). The number of supporting samples is further increased by 23% of the variants (Fig. 5b), reducing the number of SNVs unique to a single sample (Supplementary Fig. 12). Alexandrov et al.[28] determined that Signatures 1, 5, 6, and 10 are associated with CRC. While the fraction of exposures to liver cancer signatures dropped from 87% for the SNVs identified by Strelka2 to 82% for Moss (Fig. 5c), it is important to note that Moss increased the call set by more than 27%. This means that the vast majority of the newly identified SNVs are explained by known CRC signatures. We observe similar results for Patient 45, identifying an additional 24% SNVs

with an associated increase in the number of supporting samples while maintaining the total exposure to CRC mutational signatures (Supplementary Fig. 13).

## Discussion

In this work, we introduced Moss, a light-weight versatile multi-sample somatic SNV caller for bulk DNA sequencing tumor data. Moss transforms any current single-sample caller into a multi-sample caller, without having to modify the original caller's software. While two recent SNV callers, Mutect2 (GATK version 4.1) and multisnv[20], support multiple samples, the majority of methods are still single-sample only, such as Strelka2[11], MuSE[12], CaVEMan[13], VarScan[33], Lancet[14], Platypus[15], and LoFreq[16]. As such, Moss can be used in conjunction with any of these methods to improve the sensitivity of their call sets on multi-sample tumor sequencing data.

On simulated data containing $m \in \{2, \ldots, 5\}$ tumor samples, we showed that Moss increases the recall rate without losing precision in all cases, as compared to the union call set of the single-sample callers Strelka2[11] and Mutect2[10]. On real data with $m = 23$ samples from a hepatocellular carcinoma tumor[26] and $m = 3$ samples from two colorectal tumors[32], we observed that Moss recovers low-VAF variants (we verified a subset by manual review[21]) while maintaining or increasing the overall exposure to the tumor-type specific signatures. We demonstrated a similar increase in recall on $m = 2$ samples (primary tumor and relapse) of an acute myeloid leukemia dataset[22] for which a carefully curated set of somatic variants has been released. Further, we showed that the benefits of Moss are tangible even with a small number of samples or coverage as low as 10×, and come with a negligible overhead in run time and memory consumption as compared to the single-sample callers. We found that although Mutect2 (GATK version 4.1) and multisnv[20] have recently included support for the joint analysis of multiple tumor samples, Moss achieves a higher $F_1$ score on the simulated data with a shorter run time.

In summary, Moss' ability to recover low-VAF single-nucleotide variants from multi-sample bulk DNA sequencing tumor data and the resulting improvement in sensitivity will enable more detailed downstream analyses in cancer genomics. We do note that manual review of identified variants will remain a necessary step to obtain a high-quality set of variants. Moss will be most useful in a hypothesis-generating context, where the low-frequency variants identified by Moss will be excellent candidates for targeted follow-up sequencing. Due to decreasing costs of sequencing technologies, we expect multi-sample data to become increasingly available. Moss is particularly suitable to analyze liquid biopsies of tumors, which are taken at multiple time points

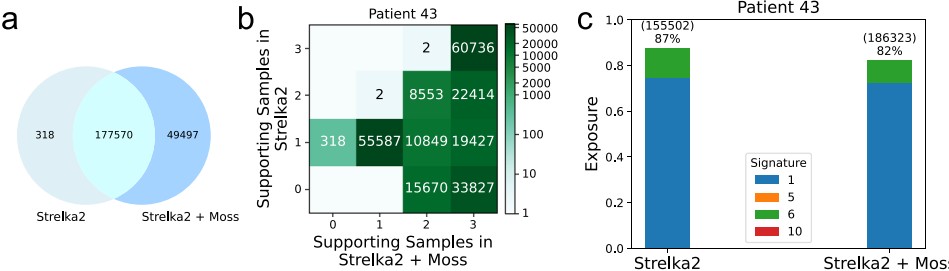

**Fig. 5 Moss recovers high-quality somatic variants missed by Strelka2 in a colorectal cancer (CRC) dataset**[32]. **a** Venn diagram comparing the call set of Strelka2 when run in isolation and when run in conjunction with Moss. Moss identifies 49,497 new variants while retaining almost all variants identified by Strelka2. **b** The number of supporting samples by Moss (x-axis) and the single-sample caller Strelka2 (y-axis) for each variant, showing that Moss increases the number of supporting samples for 23% of variants (52,690 variants) identified by Strelka2, increasing the number of common SNVs (single-nucleotide variants) that are present in at least two samples (Supplementary Fig. 12). **c** Exposure to mutational signatures of colorectal tumor for the different methods (each signature is represented with a unique color). The identified signatures remain the same.

and typically have SNVs with low VAFs due to the low prevalence of circulating tumor DNA.

There are several directions for future research. First, the concept of using multiple samples to improve variant calling sensitivity is broadly applicable beyond single-nucleotide variants. For instance, Zaccaria et al.[25] recently demonstrated that copy-number aberrations can be identified with greater accuracy when considering multiple samples from an individual tumor. We expect that the detection of small indels and larger structural variants will benefit from a similar multi-sample analysis as employed by Moss. Second, Moss is designed to work with bulk DNA sequencing tumor data, where samples are composed of a mixture of hundreds of thousands of cells. Single-cell sequencing (SCS) has recently gained increasing attention as it directly reveals the set of mutations in individual tumor cells, which can be used for tumor phylogeny reconstruction[34–37]. We plan to adapt Moss' Bayesian model to support SNV calling in SCS data. Third, as SCS data suffer from elevated error rates[38], Moss could be further extended to perform joint somatic SNV calling in hybrid tumor datasets composed of both bulk and SCS sequencing samples[39–42]. Joint variant calling on hybrid datasets will directly improve the accuracy of algorithms that perform phylogeny reconstruction on both SCS and bulk sequencing data[43,44]. Fourth, it will interesting to adapt Moss to support long-read sequencing data with increased error rates. Finally, the set of called variants could be further refined by incorporating information from cancer genomics downstream analyses. For example, Rubanova et al.[45] incorporate mutational signatures to cluster SNVs that co-occur in tumor clones. In a similar vein, the accuracy of SNV calling may improve by taking mutational signatures into account during variant calling. Another example of incorporating downstream information is the work of Singer et al.[46] who explored simultaneous SNV calling and phylogeny inference from single-cell DNA sequencing data of tumors. In the future, we plan to simultaneously call SNVs and perform cancer phylogeny inference from either hybrid datasets or multi-sample bulk DNA data of tumors.

## Methods

Moss is a multi-sample single-nucleotide variant (SNV) caller that is applied after running a single-sample variant caller independently in each sample (Fig. 1b). In addition to the original BAM files of the tumor and normal samples, Moss optionally takes as input the realigned BAM files and the candidate loci. The candidate loci are taken as the union of the reported loci in the VCF files output by the single-sample caller under permissive conditions. The output of Moss is a VCF file containing the set of called variants. The realigned BAM files are paired with the original BAM files, and for each sample, the modified reads in the original BAM file are updated with the realigned ones.

Here, we present the model of Moss, which is a Bayesian model for inferring the posterior probability of whether a location in the genome contains a somatic SNV (single-nucleotide variant) and the corresponding tumor allele. In the following, we will refer to the posterior probability as the somatic probability, and we will focus on a specific locus in the genome to describe the proposed method Moss, as the model treats different loci independently.

**Graphical model**. Our algorithm Moss may be summarized as a graphical model, which is shown in Fig. 6. For a sample $i$ among a total of $m$ samples, we denote by $n_i$ the number of aligned reads that span the locus in question. We use $\mathbf{b}_i = [b_{i,1}, \ldots, b_{i,n_i}]$ and $\mathbf{q}_i = [q_{i,1}, \ldots, q_{i,n_i}]$ to represent the set of bases and the corresponding quality scores of the reads, respectively. We denote the allele in the normal sample by $\mathcal{N}$, inferred by the used single-sample SNV caller and taken from the corresponding VCF file (this is an input to Moss). In line with current somatic SNV callers[10,15], we only consider the case where the normal sample at the considered locus is homozygous (most usual case) and ignore heterozygous loci. For the somatic SNVs, we only allow one somatic allele, and we use $\mathcal{T}$ to represent the latent mutated allele of the SNV, which is by definition different from $\mathcal{N}$. For example, if the normal sample contains nucleotide "A" in the considered locus and the tumor sample introduces the somatic SNV "A > G", then we have $\mathcal{N} = $ A and $\mathcal{T} = $ G.

We also define the error-free bases $\mathbf{x}_i = [x_{i,1}, \ldots, x_{i,n_i}]$ corresponding to the observed bases $\mathbf{b}_i$, and $f_i$ as the latent variable representing the somatic variant allele

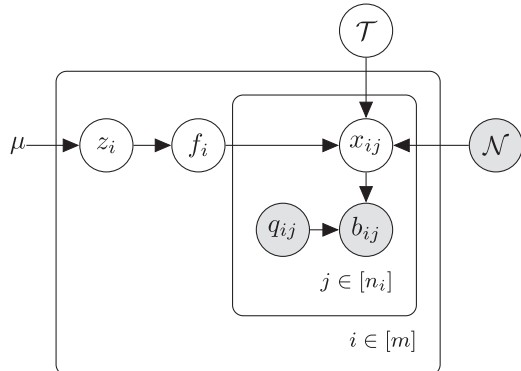

**Fig. 6 Graphical model used for the computation of the likelihood of a locus across multiple samples having a somatic SNV (single-nucleotide variant), as well as the corresponding tumor allele $\mathcal{T}$.** Different loci are analyzed independently. We consider $m$ samples, with $n_i$ reads for each sample mapping to the considered locus. $\mu$ is the somatic mutation rate; $z_i$ is the latent binary indicator for sample $i$ carrying the somatic SNV; $\mathcal{T}$ and $\mathcal{N}$ are the tumor and normal allele, respectively; $x_{ij}$ is the error-free base; $b_{ij}$ is the observed base; and $q_{ij}$ is the sequencing error probability.

frequency in sample $i$. Hence, variable $x_{ij}$ is either $\mathcal{N}$ or $\mathcal{T}$, and it follows a Bernoulli distribution with probability $f_i$ of being equal to $\mathcal{T}$. Each base $b_{ij}$ in $\mathbf{b}_i$ (for sample $i$) is assumed to be generated following a Categorical distribution parameterized by the true base $x_{ij}$ and the sequencing error rate $q_{ij}$, modeling whether there was a sequencing error or not. Hence, conditioned on $\mathbf{x}_i$ and $\mathbf{q}_i$, $\mathbf{b}_i$ is modeled as a Multinomial distribution. The distribution of the frequency $f_i$ depends on the latent binary variable $z_i$, which indicates whether sample $i$ carries the somatic SNV or not. The prior distribution of $\mathbf{z} = [z_1, \cdots, z_m]$ depends on the parameter $\mu$, which indicates the probability of having a somatic SNV in a given locus. With the proposed model, the bases of each sample are generated independently conditioned on $\mathbf{z}$, $\mathcal{N}$, and $\mathcal{T}$.

**Somatic probability**. We seek to compute the somatic probability $\mathbf{P}(\mathbf{Z} \neq \mathbf{0} \mid \mathcal{N}, \mathbf{b}, \mathbf{q})$, which indicates the probability of having at least one sample containing a somatic variant at the considered locus given the normal allele $\mathcal{N}$ and the set $\mathbf{b}$ of bases and quality scores $\mathbf{q}$ from the $m$ samples that map to the considered locus in the genome. To improve efficiency, we compute the complement of the somatic probability instead. That is, we compute $\mathbf{P}(\mathbf{Z} = \mathbf{0} \mid \mathcal{N}, \mathbf{b}, \mathbf{q}) = 1 - \mathbf{P}(\mathbf{Z} \neq \mathbf{0} \mid \mathcal{N}, \mathbf{b}, \mathbf{q})$ as

$$\mathbf{P}(\mathbf{Z} = \mathbf{0} \mid \mathcal{N}, \mathbf{b}, \mathbf{q}) = \frac{\mathbf{P}(\mathbf{b} \mid \mathbf{Z} = \mathbf{0}, \mathcal{N}, \mathbf{q})\mathbf{P}(\mathbf{Z} = \mathbf{0})}{\sum_{\mathcal{T} \neq \mathcal{N}}\sum_{\mathbf{z} \in \{0,1\}^m}\mathbf{P}(\mathbf{b}, \mathbf{Z} = \mathbf{z}, \mathcal{T} \mid \mathcal{N}, \mathbf{q})}$$
$$= \frac{\sum_{\mathcal{T} \neq \mathcal{N}}\mathbf{P}(\mathbf{b} \mid \mathbf{Z} = \mathbf{0}, \mathcal{T}, \mathcal{N}, \mathbf{q})\mathbf{P}(\mathbf{Z} = \mathbf{0})}{\sum_{\mathcal{T} \neq \mathcal{N}}\sum_{\mathbf{z} \in \{0,1\}^m}\mathbf{P}(\mathbf{b} \mid \mathbf{Z} = \mathbf{z}, \mathcal{T}, \mathcal{N}, \mathbf{q})\mathbf{P}(\mathbf{Z} = \mathbf{z})} \quad (1)$$

where $\mathbf{b} = \{\mathbf{b}_1, \cdots, \mathbf{b}_m\}$ denotes all bases that aligned to the considered locus in all $m$ samples, with $\mathbf{b}_i$ representing the bases in sample $i$. Analogous notation is used for $\mathbf{q}$, which represents the quality scores. To derive Eq. (1), we have used the fact that the somatic indicator $\mathbf{z}$ and the tumor genotype $\mathcal{T}$ are mutually independent and independent of $\mathcal{N}$ (see also Fig. 6).

We assume the prior of $\mathbf{z}$ to be uniform across all cases where $\mathbf{z} \neq \mathbf{0}$, and equal to $1 - \mu$ for $\mathbf{z} = \mathbf{0}$. In particular, we have

$$\mathbf{P}(\mathbf{z}) = \begin{cases} 1 - \mu, & \text{if } \mathbf{z} = \mathbf{0}, \\ \frac{\mu}{2^m - 1}, & \text{if } \mathbf{z} \neq \mathbf{0}. \end{cases} \quad (2)$$

The likelihoods of the bases of the $m$ samples are conditionally independent given $\mathbf{z}, \mathcal{T}, \mathcal{N}$, which leads to

$$\mathbf{P}(\mathbf{b} \mid \mathbf{Z} = \mathbf{z}, \mathcal{T}, \mathcal{N}, \mathbf{q}) = \prod_{i=1}^{m} \mathbf{P}(\mathbf{b}_i \mid Z_i = z_i, \mathcal{T}, \mathcal{N}, \mathbf{q}_i) . \quad (3)$$

The sample-specific data likelihood is computed by marginalizing the allele frequency $f_i$ as follows.

$$\mathbf{P}(\mathbf{b}_i \mid Z_i = z_i, \mathcal{T}, \mathcal{N}, \mathbf{q}) = \int_0^1 \mathbf{P}(\mathbf{b}_i, f_i \mid z_i, \mathcal{T}, \mathcal{N}, \mathbf{q}_i)\mathrm{d}f_i$$
$$= \int_0^1 \mathbf{P}(\mathbf{b}_i \mid f_i, \mathcal{T}, \mathcal{N}, \mathbf{q}_i)\mathbf{P}(f_i \mid z_i)\mathrm{d}f_i \quad (4)$$

In Eq. (4) we have used the fact that conditioned on $f_i$, $\mathbf{b}_i$ is independent of $z_i$. Conditioned on $f_i, \mathcal{T}, \mathcal{N}, \mathbf{q}_i$, and assuming no sequencing errors, the likelihood of

each base $b_{ij}$ of sample $i$ being equal to the tumor allele $\mathcal{T}$ follows a Bernoulli distribution with parameter $f_i$. Hence, the likelihood $\mathbf{P}(\mathbf{b}_i \mid f_i, \mathcal{T}, \mathcal{N}, \mathbf{q}_i)$ of observing $d_{t,i}$ tumor bases out of a total of $n_i$ bases is given by $\binom{d_{n,i} + d_{t,i}}{d_{n,i}}(1 - f_i)^{d_{n,i}} f_i^{d_{t,i}}$, where the number of normal bases $d_{n,i}$ equals $n_i - d_{t,i}$. To take sequencing errors into account, we introduce for each sample $i$ variable $\mathbf{x}_i = [x_{i1}, \ldots, x_{in_i}]$, with $x_{ij}$ representing the true base $j$ of sample $i$. Hence, $b_{ij} \neq x_{ij}$ indicates a sequencing error. Note that in our formulation $x_{ij}$ is either $\mathcal{N}$ or $\mathcal{T}$. Hence, conditioned on $f_i, \mathcal{T}$ and $\mathcal{N}$, $x_{ij}$ follows a Bernoulli distribution.

$$\mathbf{P}\left(x_{ij} \mid f_i, \mathcal{T}, \mathcal{N}\right) = \begin{cases} f_i, & \text{if } x_{ij} = \mathcal{T}, \\ 1 - f_i, & \text{if } x_{ij} = \mathcal{N}. \end{cases} \quad (5)$$

We further assume that a sequencing error is not biased towards a particular base (a similar assumption is made in the derivation of Strelka2[11]). Therefore, the likelihood of base $b_{ij}$ conditioned on $x_{ij}$ and $q_{ij}$ is given by

$$\mathbf{P}\left(b_{ij} \mid x_{ij}, q_{ij}\right) = \begin{cases} 1 - q_{ij}, & \text{if } b_{ij} = x_{ij}, \\ q_{ij}/3, & \text{otherwise}. \end{cases} \quad (6)$$

Finally, combining Eqs. (5) and (6), the data likelihood of sample $i$ can be computed as:

$$\begin{aligned}\mathbf{P}(\mathbf{b}_i \mid f_i, \mathcal{T}, \mathcal{N}, \mathbf{q}_i) &= \rho_i \prod_{j=1}^{n_i} \mathbf{P}\left(b_{ij} \mid f_i, \mathcal{T}, \mathcal{N}, q_{ij}\right) \\ &= \rho_i \prod_{j=1}^{n_i} \sum_{x_{ij} = \mathcal{N}, \mathcal{T}} \mathbf{P}\left(b_{ij} \mid q_{ij}, x_{ij}\right) \mathbf{P}\left(x_{ij} \mid f_i, \mathcal{T}, \mathcal{N}\right),\end{aligned} \quad (7)$$

where $\rho_i = \binom{d_{n,i} + d_{t,i} + d_{e,i}}{d_{n,i}, d_{t,i}, d_{e,i}}$ is the multinomial coefficient for sample $i$, $d_{e,i}$ is the number of error bases $\left(b_{ij} \notin \{\mathcal{N}, \mathcal{T}\}\right)$, and $d_{n,i} + d_{t,i} + d_{e,i} = n_i$.

Next, we specify how $\mathbf{P}(f_i \mid z_i)$ is computed. Without further prior knowledge on the somatic SNV frequency $f_i$, a uniform prior is used when $z_i = 1$, i.e., when a somatic SNV is present. Similarly to Strelka2[11], we assume a uniform prior on the range $[\epsilon, 1]$ for $f_i$ where $\epsilon$ is set to 0.05 by default. If the somatic SNV is absent, i.e., if $z_i = 0$, the distribution of $f_i$ should be concentrated around 0. For simplicity, we force $f_i$ to be zero in this case. To summarize, we have:

$$\mathbf{P}\left(f_i \mid z_i = 1\right) = \begin{cases} \frac{1}{1-\epsilon}, & \text{if } f_i \in [\epsilon, 1], \\ 0, & \text{otherwise}, \end{cases} \quad (8)$$

and

$$\mathbf{P}\left(f_i \mid z_i = 0\right) = \delta\left(f_i\right), \quad (9)$$

where $\delta()$ is the Dirac delta function.

Note that if $z_i = 0$, then $f_i = 0$ and the likelihood can be simplified to:

$$\mathbf{P}(\mathbf{b}_i \mid Z_i = 0, \mathcal{T}, \mathcal{N}, \mathbf{q}_i) = \rho_i \prod_{j=1}^{|\mathbf{b}_i|} \left(1 - q_{ij}\right)^{\mathbb{1}\{b_{ij} = \mathcal{N}\}} \cdot \left(\frac{q_{ij}}{3}\right)^{\mathbb{1}\{b_{ij} \neq \mathcal{N}\}}, \quad (10)$$

which is independent of $\mathcal{T}$.

**Estimation of the tumor allele $\mathcal{T}$.** To estimate the tumor allele of the somatic SNV, we use maximum likelihood (ML) to estimate the allele $\mathcal{T}$ that minimizes

$$\mathbf{P}(\mathbf{Z} = \mathbf{0} \mid \mathbf{b}, \mathcal{T}, \mathcal{N}, \mathbf{q}) = \frac{\mathbf{P}(\mathbf{b} \mid \mathbf{Z} = \mathbf{0}, \mathcal{T}, \mathcal{N}, \mathbf{q})\mathbf{P}(\mathbf{Z} = \mathbf{0})}{\sum_{\mathbf{z} \in \{0,1\}^m} \mathbf{P}(\mathbf{Z} = \mathbf{z}, \mathbf{b}, \mathcal{T} \mid \mathcal{N}, \mathbf{q})} \quad (11)$$

In this special case, based on Eq. (10) and the definitions of priors for $\mathbf{Z}$ and $\mathcal{T}$, the numerator $\mathbf{P}(\mathbf{b} \mid \mathbf{Z} = \mathbf{0}, \mathcal{T}, \mathcal{N}, \mathbf{q})\mathbf{P}(\mathbf{Z} = \mathbf{0})$ is a constant independent of $\mathcal{T}$. Therefore, we may estimate $\mathcal{T}$ as

$$\mathcal{T}^* = \arg\max_{\mathcal{T}} \sum_{\mathbf{z} \in \{0,1\}^m} \mathbf{P}(\mathbf{b} \mid \mathbf{z}, \mathcal{T}, \mathcal{N}, \mathbf{q})\mathbf{P}(\mathbf{z}), \quad (12)$$

which is already calculated in the denominator of Eq. (1) for computing the somatic probability. Note that, if only one tumor sample contains reads with $\mathcal{T}$, we defer to the single-sample caller and call the SNV only if it was called by the original caller.

**Omission of samples.** By default, Moss ignores samples in which all bases aligned to the considered locus are equal to the normal allele $\mathcal{N}$ (i.e., all-normal samples), as we can show that the somatic probability diminish to 0 as the number of all-normal samples increases. A detailed explanation can be found in the supplement.

**Filters.** Moss filters out reads with a mapping quality below 30 and bases with quality below 13 from the BAM file before performing variant calling. Note that this filtering criteria was also used in[26]. In order to detect possible false positive variants, Moss supports several additional filters based on information provided in the VCF file. In particular, Moss applies the following empirical filters described in[26]: (i) low-normal-depth, for variants with depth of normal sample less than 6

(DP field), (ii) low-tumor-support, for variants whose tumor allele count is less than 4 in all samples (TCOUNT field), and (iii) empty-strand, for variants with no allele counts on the forward or reverse strand in all samples (SB field). Moreover, Moss is equipped with an additional cluster filter for clustered SNV sites (at least 3 SNV sites within 100 base-pairs), as they are likely to be artefacts caused by undetected duplicated reads. By default, Moss applies these filters only to variants discovered by Moss that were undetected by the single-sample variant tool used prior to Moss. Finally, we have a separate filtering criterion to detect normal contamination that we describe in the following.

**Filtering contamination.** In some cases, the tumor allele $\mathcal{T}$ is present at a high frequency in the normal sample. This may be due to either contamination of the normal sample or due to an inaccurate identification of the normal allele $\mathcal{N}$. To arrive at a set of high-quality variants, one may wish to identify these variants and subsequently filter them out. One approach could consist on applying a hard threshold on the tumor allele counts present in the normal sample. However, a hard threshold does not account for varying sequence samples characteristics such as the coverage, which can have an impact on which variants are filtered out. Hence, here we provide another score, tumor alleles in normal (TIN), which represents the posterior of the normal sample carrying the previously inferred tumor allele $\mathcal{T}$. In particular, the TIN score is given by $-10 \log \mathbf{P}(Z_0 = 1 \mid \mathbf{b}_0, \mathbf{b}, \mathcal{T}, \mathcal{N}, \mathbf{q})$. This score can be calculated alongside the somatic SNV probability with little overhead as follows:

$$\mathbf{P}(Z_0 = 1 \mid \mathbf{b}_0, \mathbf{b}, \mathcal{T}, \mathcal{N}, \mathbf{q}) = \frac{\sum_{\mathbf{z} \in \{0,1\}^m} \mathbf{P}(\mathbf{b}_0, \mathbf{b} \mid Z_0 = 1, \mathbf{z}, \mathcal{T}, \mathcal{N}, \mathbf{q})\mathbf{P}(Z_0 = 1, \mathbf{z})}{\sum_{\mathbf{z} \in \{0,1\}^m, Z_0 \in \{0,1\}} \left(\mathbf{P}(\mathbf{b}_0, \mathbf{b} \mid Z_0, \mathbf{z}, \mathcal{T}, \mathcal{N}, \mathbf{q})\mathbf{P}(Z_0, \mathbf{z})\right)}, \quad (13)$$

where subscript 0 represents the normal sample. By default, variants with TIN score below 20 are filtered out (this threshold can be modified by the user). Similarly to the empirical filters, the TIN score is applied only to the somatic variants discovered by Moss that were undetected by the single-sample somatic variant caller applied prior to Moss.

**Efficient calculation of the somatic probability.** Calculating the denominator of Eq. (1) requires summing over all possible $2^m$ combinations of $\mathbf{z}$, which gives an exponential complexity of $O(2^m)$ and hence does not scale with the number of samples. However, given the considered prior $\mathbf{P}(\mathbf{z})$ (see Eq. (2)), we can rewrite it in a simpler form with linear complexity $O(m)$:

$$\begin{aligned}&\sum_{\mathbf{z} \in \{0,1\}^m} \mathbf{P}(\mathbf{b} \mid \mathbf{Z} = \mathbf{z}, \mathcal{T}, \mathcal{N}, \mathbf{q})\mathbf{P}(\mathbf{Z} = \mathbf{z}) \\ &= \left(1 - \frac{2^m}{2^m - 1}\mu\right) \prod_{i=1}^{m} \mathbf{P}(\mathbf{b}_i \mid z_i = 0, \mathcal{T}, \mathcal{N}, \mathbf{q}) \\ &\frac{\mu}{2^m - 1} \prod_{i=1}^{m} \left(\mathbf{P}(\mathbf{b}_i \mid z_i = 0, \mathcal{T}, \mathcal{N}, \mathbf{q}) + \mathbf{P}(\mathbf{b}_i \mid z_i = 1, \mathcal{T}, \mathcal{N}, \mathbf{q})\right)\end{aligned} \quad (14)$$

**Experimental details.** The analyses performed in this study included bulk DNA sequencing data (see "Data availability") in the form of FASTQ (for the simulated and HCC samples) and BAM/SAM files (for the AML and CRC samples). To obtain our simulated data, we used MASCoTE[25] to generate the BAM files, which internally uses ART (v2.5.8) for simulating sequencing reads and BWA (v0.7.17) for alignment. For the HCC dataset, we generated the BAM files by aligning FASTQ files to the human reference genome b37 using BWA-MEM (v0.7.17) with parameters -t 16 -M. For the CRC dataset, we used BWA-MEM (v0.7.17) with parameters -t 14 -M -Y to align reads in FASTQ files to the human reference genome GRCh38. The aligned BAM files of the HCC and CRC samples were further processed with GATK (v4.0.12) following GATK's best practices (including BQSR and MarkDuplicates). For the AML dataset, we directly used the BAM files downloaded from dbGaP.

Single sample callers Mutect2 from GATK (v4.0.12) and Strelka2 (v2.9.9) were used to identify SNVs, as well as multi-sample callers Mutect2 from GATK (v4.1.7.) and multisnv (v2.3-14-gb86d4dc). To calculate the exposure to COSMIC v2 mutational signatures we used deconstructSig (v1.8.0). We performed manual review using IGV (v2.4.16) and the add-on IGVNav[21] (commit 8df35a6) and followed their recommended standard operating procedures (SOP). Specifically, each time we examined three BAM files at the same time: the normal sample, the original BAM file, and the realigned BAM file. We assigned tags and notes according to the SOP and used these tags to call and classify the variant.

**Reporting summary.** Further information on research design is available in the Nature Research Reporting Summary linked to this article.

## Data availability

The generated simulated data used in this study is available at figshare [https://doi.org/10.6084/m9.figshare.14079953]. The analyzed HCC data[26] is available at genome sequence archive of Beijing Institute of Genomics under accession id PRJCA000091. The AML data[22] is available at the database of Genotypes and Phenotypes (dbGaP) under accession id dbGaP:phs000159. The CRC data[32] is available under restricted access,

which can be obtained upon signing a Data Use Agreement by contacting Nicholas Chia [Mayo Clinic, Rochester, MN, USA, Chia.Nicholas@mayo.edu]. Data are available within the Article, Supplementary Information or available from the authors upon request. Source data are provided with this paper.

## Code availability

Moss is open source (MIT license) and available at https://github.com/elkebir-group/Moss and on Zenodo https://doi.org/10.5281/zenodo.4487204[47]. In addition, Moss is available on conda and as a Docker image. The installation and usage guide are in the last section of the Supplementary Information.

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

## Acknowledgements

M.E.K. was supported by the National Science Foundation (grant: CCF 1850502). I.O. was partially supported by a Gipuzkoa Fellows grant from the Basque Government. We thank Nicholas Chia for providing access to the CRC data.

## Author contributions

M.E.K. and I.O. conceived the project. All the authors contributed to the development of the algorithm and C.Z. implemented the software. M.E.K. and I.O. led interpretation of the experimental results. All authors contributed to the writing of the manuscript. M.E.K. and I.O. supervised the project.

## Competing interests

The authors declare no competing interests.

**Additional information**

