## [Peer Review File · Nature Communications]

REVIEWER COMMENTS

Reviewer #1 (Remarks to the Author): Expert in cancer genomics

The manuscript "Moss: Accurate Single-Nucleotide Variant Calling from Multiple Bulk DNA Tumor Samples" by Zhang et al. describes a mathematical approach to improve on SNV calling in tumor-normal comparisons. The premise is that joining the analysis of multiple comparisons within a Bayesian framework beats somatic mutation calling using a 1:1 approach with commonly used softwares such as Mutect2 or Strelka. In principle this is interesting and the execution seems to be good. However, there are several flaws in the premise: 1) it is still uncommon to carry out multiple independent samplings of the same tumor and sequencing them independently and then comparing them to a corresponding normal samples. Commonly, only one tumor sample is sequenced. Cost is still prohibitive to analyze multiple tumor samples. 2) Different types of cancer pose very different scenarios, many are reasonably homogeneous and identifying somatic mutations that appeared late can be interesting. For this Moss could be interesting. Others, such as prostate cancer, can be multifocal where multiple sampling would make sense, but on the other hand very different somatic mutation profiles can be obtained from different locations. For such a situation this algorithm would quite likely not help a lot. 3) The emphasis is on SNV calling, while other types of somatic mutations, such as insertion/deletions are also of great interest. They generally pose a far bigger challenge to identify robustly.

Overall this work is very mathematical/computational and probably would receive more appreciation in a journal specialized in computational topics.

Reviewer #2 (Remarks to the Author): Expert in bioinformatics, SNV calling and analysis, and cancer heterogeneity

The Moss tool takes single-sample somatic mutation calls from individual tumors and extends them across multiple tumor samples from the same individual. This is potentially quite useful because variants at high variant allele frequencies in one sample may be at low VAFs, and thus uncalled, in other spatially or temporally distinct samples. Even more interesting are mutations that are at low VAFs in multiple samples, but are often filtered out for having low levels of evidence. Combining data across samples is potentially a powerful method for detecting these.

The method appears sound, and the recognition of key "real-world" factors like tumor contamination in normal samples is welcome. Overall, the paper is well-written and method well done.

Simulated data is appropriately used, as there's no way to have "ground truth" from an actual tumor sample. The simulation experiments are solid. My primary concern comes from the real tumor data.

The increased recall rate in these samples is evident, and impressive, but the assessment of precision is less convincing. While observing similar mutational patterns certainly provides some evidence that the new mutations are real, that seems insufficient, especially when dealing with

mutations that are at low VAF in every sample.

To put it another way, events that recur in multiple samples are either really interesting, because they are shared mutations, or really bad, because they represent recurrent artifacts (often in regions of the genome that are misassembled or prone to sequencing errors).

Expert manual review still represents the gold standard for evaluating whether mutations are truly somatic, and detailed guides to doing so have been published: (see <https://www.nature.com/articles/s41436-018-0278-z>, especially the supplement). I would suggest that the authors take at least a fraction of the new calls from the real tumor samples and manually review them to determine whether there is evidence that they are artifacts, to the best of their ability to discern.

Another way to address the issue of precision could be to use the data in [https://www.cell.com/fulltext/S2405-4712\(15\)00113-1](https://www.cell.com/fulltext/S2405-4712(15)00113-1), where a tumor and relapse pair were sequenced to very high depths (allowing for detection of very low VAF events with high confidence), and tens of thousands of variants were manually reviewed. This is the closest thing to a "gold standard" that I'm aware of for tumor sequencing data. This "pre-reviewed" data could provide another high quality data set (though only with two samples) for establishing precision.

Other notes/questions of lesser importance:

- It would be quite interesting to see what happens if you ran variant callers like strelka/mutect with "relaxed" parameters (rather than default params), then apply Moss. Does that allow for detection of high quality low-VAF variants that would have otherwise been absent from all input VCFs (and thus never made it into Moss)? Evidence that this works (with reasonable precision) would make it an even more powerful tool.

- The data sets are fairly low coverage by modern standards (though I didn't see coverage info for the CRC samples). Exomes are often sequenced to 200-300x, especially when heterogeneity is being studied. (I definitely understand that samples with 23 biopsies are a unique and rare resource, but tumors with 2-5 samples and higher depth sequencing are plentiful). The core question is: How does deeper coverage data (with potentially more true and false positives at low VAFs) affect the ability of Moss to accurately infer true somatic events? In essence, this question is asking "how low can you go?"

- The software is well documented and has a permissive license, which is appreciated. My only note about code is that a docker container would go a long way towards making your software more easily usable in modern sequencing pipelines!

- Is this model generalizable enough to cover small indels as well? That extension certainly isn't required for this publication, but would be interesting to see.

Point-by-point Response to Reviewers

Nature Communications manuscript NCOMMS-20-22872

We thank the reviewers for their thoughtful comments. We believe that this new submitted version addresses all the reviewers' comments. In particular, following Reviewer #2's suggestions, we included additional validation experiments to assess precision in the following two ways. First, we performed a manual review process on the HCC dataset. Second, we analyzed an additional high-coverage AML dataset with an established high-quality gold list of SNVs. These two experiments demonstrate that Moss' multi-sample analysis improves recall while retaining high precision compared to current single-sample SNV callers. Moreover, we updated our Introduction and Discussion following Reviewer #1's comments. A detailed point-by-point response is provided below.

Reviewer #1

Comment 1.1 The manuscript "Moss: Accurate Single-Nucleotide Variant Calling from Multiple Bulk DNA Tumor Samples" by Zhang et al. describes a mathematical approach to improve on SNV calling in tumor-normal comparisons. The premise is that joining the analysis of multiple comparisons within a Bayesian framework beats somatic mutation calling using a 1:1 approach with commonly used softwares such as Mutect2 or Strelka. In principle this is interesting and the execution seems to be good. However, there are several flaws in the premise: 1) it is still uncommon to carry out multiple independent samplings of the same tumor and sequencing them independently and then comparing them to a corresponding normal samples. Commonly, only one tumor sample is sequenced. Cost is still prohibitive to analyze multiple tumor samples.

Response: Multiple samples are essential to better characterize and study intra-tumor heterogeneity. Compared to bulk DNA sequencing of a single tumor sample along with a matched normal sample, multi-sample datasets enable more precise deconvolution of mutations into clones and ultimately a more precise reconstruction of a tumor's evolutionary history. This is increasingly recognized in the community and since Gerlinger et al.'s seminal paper, multi-sample datasets are increasingly being generated. We expect this trend to continue, especially in light of profiling using liquid biopsies as well as further decreasing costs in sequencing. However, SNV callers have not caught up yet and are still geared toward single-sample datasets, a limitation that this manuscript addresses. We have made this point more clear in the introduction, where we now write:

"Importantly, the aforementioned variant callers take as input a single tumor sample only. However, multi-sample datasets enable a more precise characterization of the clones present in a tumor as well as the tumor's evolutionary history. This, coupled with decreasing sequencing costs and the availability of new profiling techniques such as liquid biopsies, have led to an increasing availability of multi-sample data. Current single-sample SNV callers are unable to unlock the potential of these data, which enable more accurate variant calling because the probability of the same sequencing error occurring in all tumor samples at the same locus decreases significantly with an increasing number of tumor samples (Fig. 1a)."

Comment 1.2 2) Different types of cancer pose very different scenarios, many are reasonably homogeneous and identifying somatic mutations that appeared late can be interesting. For this Moss could be interesting. Others, such as prostate cancer, can be multi-focal where multiple sampling would make sense, but on the

other hand very different somatic mutation profiles can be obtained from different locations. For such a situation this algorithm would quite likely not help a lot.

Response: We agree that the benefit of Moss will vary depending on the tumor type and the resulting variability between samples. Nevertheless, we have shown that for all analyzed data, running Moss in conjunction with a single-sample caller increases the number of supporting samples for a large proportion of the common variants identified by both methods. Additionally, in the revised version of the manuscript, we explicitly show that Moss reduces the number of common variants that are present in only one sample (spatial samples for CRC and HCC, temporal samples for AML). Thus, we find that the use of a single-sample SNV caller will overstate the diversity in temporal/spatial distribution of SNVs (Supplementary Figures S5, S11, and S12).

Comment 1.3 3) The emphasis is on SNV calling, while other types of somatic mutations, such as insertion/deletions are also of great interest. They generally pose a far bigger challenge to identify robustly.

Response: Reviewer #2 made the same point. We agree that is a valuable future direction of research, which we now list in the Discussion:

“There are several directions for future research. First, the concept of using multiple samples to improve variant calling accuracy is broadly applicable beyond single-nucleotide variants. For instance, Zaccaria et al. recently demonstrated that copy number aberrations can be identified with greater accuracy when considering multiple samples from an individual tumor. We expect that the detection of small indels and larger structural variants will benefit from a similar multi-sample analysis as employed by Moss.”

Comment 1.4 Overall this work is very mathematical/computational and probably would receive more appreciation in a journal specialized in computational topics.

Response: We believe that this work is suitable for this journal, which has a history of publishing computational/methods papers. We leave the decision to the editor.

Reviewer #2

Comment 2.1 The Moss tool takes single-sample somatic mutation calls from individual tumors and extends them across multiple tumor samples from the same individual. This is potentially quite useful because variants at high variant allele frequencies in one sample may be at low VAFs, and thus uncalled, in other spatially or temporally distinct samples. Even more interesting are mutations that are at low VAFs in multiple samples, but are often filtered out for having low levels of evidence. Combining data across samples is potentially a powerful method for detecting these.

The method appears sound, and the recognition of key “real-world” factors like tumor contamination in normal samples is welcome. Overall, the paper is well-written and method well done.

Simulated data is appropriately used, as there’s no way to have “ground truth” from an actual tumor sample. The simulation experiments are solid. My primary concern comes from the real tumor data.

The increased recall rate in these samples is evident, and impressive, but the assessment of precision is less convincing. While observing similar mutational patterns certainly provides some evidence that the new mutations are real, that seems insufficient, especially when dealing with mutations that are at low VAF in every sample.

To put it another way, events that recur in multiple samples are either really interesting, because they are shared mutations, or really bad, because they represent recurrent artifacts (often in regions of the genome that are misassembled or prone to sequencing errors).

Expert manual review still represents the gold standard for evaluating whether mutations are truly somatic, and detailed guides to doing so have been published: (see <https://www.nature.com/articles/s41436-018-0278-z>, especially the supplement). I would suggest that the authors take at least a fraction of the new calls from the real tumor samples and manually review them to determine whether there is evidence that they are artifacts, to the best of their ability to discern.

Response: We thank the reviewer for positive comments and the suggestion. We have performed manual review following the suggested protocol on a subset of the variants identified by Moss on the HCC dataset (with $m = 23$ samples) when run in conjunction with Mutect2. Specifically, we have analyzed all variants called by Moss in 2 samples (before applying any filters) but not discovered by Mutect2 in any sample. There are 166 such variants. When analyzing a given variant, we look at the specific locus in both samples in isolation, and label them in each sample as Somatic (S), Germline (G), Ambiguous (A) (for variants that look correct) or Failed (F) following the manual review protocol. Since these variants all had low VAF in the corresponding tumor samples, they could not be designated as ‘S’ following the manual review criteria. None of the 166 variants were designated as ‘G’. While variants indicated as ‘F’ are sequencing/mapping artifacts, variants indicated as ‘A’ lack strong evidence to be classified as either a correct variant or an artifact. Importantly, variants labeled as ‘A’ in both samples are likely correct. This in line with our premise that the presence of low-frequency variants in multiple samples enables to distinguish them from sequencing errors. We also annotate each locus with the specific tags and comments if applicable (a detailed table with all annotations has been provided as Supplementary Table 1).

Out of the 166 variants, 19 were filtered out by Moss. Manual review of these 19 filtered variants corroborated Moss’ filtering criteria (Figure 1a and 1b). Moreover, manual review of the remaining $166 - 19 = 147$ variants resulted in 102 variants annotated as (A,A), thus corresponding to correct variants (Figure 1b). Further analysis of the variants that failed in at least one sample, as well as some randomly selected additional variants, revealed that most of them failed because of being in a high-discrepancy region (HDR) or multiple mismatches (MM) region.

Based on these results, we have added a *cluster* filter in the revised version of Moss to detect and filter HDR/MM events (defined as 3 sites containing SNVs within 100bp distance). After applying the cluster filter, Moss retains 145 SNVs (which locate at (2, 0) square of the Fig. 3c in main text), which contain the 102 SNVs annotated as (A,A) and 32 SNVs annotated as ambiguous in at least one sample (Figure 1c). When analyzing all variants, applying the *cluster* filter removed in total 1,393 SNVs that were unique to Moss (see Fig. II).

All results in the revised version of the manuscript have been updated with the latest version of Moss, as well as the GitHub repository. We have also added a paragraph in the main manuscript about the performed manual review, and referred to the added Supplementary Table 1. Specifically, we have included the following:

“For further validation, we perform manual review of the variants called by Moss in exactly two samples but not called by Mutect2 in any sample. We follow the procedure of Barnell et al. Prior to applying Moss’ filtering criteria there are 166 such variants (Supplementary Fig. 7a). After filtering (described in Methods), Moss identifies 145 variants. Manual review suggests that 102 of these are true SNVs (Fig. 3f, Supplementary Fig. 7b, and Supplementary Table 1). Of the remaining $145 - 102 = 43$ SNVs called by Moss, 32 were identified as ambiguous in at least one sample. As for the $166 - 145 = 21$ variants filtered out by Moss, 1 SNV is flagged as tumor-in-normal (TIN), 8 SNVs as empty-strand, 10 as low-tumor-support, and 7 as cluster (note that a SNV can have more than one flag). Manual review of these variants results in assigned tags that match Moss’ filtering criteria (Supplementary Table 1). When analyzing all variants, the TIN filter flagged in total 183 SNVs, empty-strand 445 SNVs, low-tumor-support 772 and cluster 701 SNVs. These findings demonstrate that the implemented filters are capable of removing artifacts, and that

the majority of the analyzed SNVs newly called by Moss pass manual inspection.”

Figure I: Annotations of a) all variants called by Moss in 2 samples and not identified by Mutect2 in any sample prior to filtering, b) retained variants after applying filters *TIN*, *low-normal-depth*, *low-tumor-support* and *empty-strand*, and c) retained variants after additionally applying the *cluster* filter.

Figure II: Changes in number of called variants after applying the *cluster* filter in Moss in the HCC data (in conjunction with Mutect2).

Finally, to illustrate some examples of removed and retained SNV sites for these data, we enclose some figures of the performed manual review (Fig. III). In the first two examples (Fig. IIIa and IIIb), the removed SNVs are in a highly discrepancy region (HDR) where nearby clustered variants show up in the same read repeatedly. In the third example (Fig. IIIc), SNVs are both present in 2 samples and there are 2 variant allele reads in each sample so that we label it as ambiguous in both samples.

Comment 2.2 Another way to address the issue of precision could be to use the data in [https://www.cell.com/fulltext/S2405-4712\(15\)00113-1](https://www.cell.com/fulltext/S2405-4712(15)00113-1), where a tumor and relapse pair were sequenced to very high depths (allowing for detection of very low VAF events with high confidence), and tens of thousands

(a) Failed because of “HDR” in sample Z1.

(b) Failed because of “HDR” in sample Z1.

(c) Ambiguous SNV in sample B5 and C2. Identified by Moss as SNV.

Figure III: Three examples of the performed manual review.

of variants were manually reviewed. This is the closest thing to a "gold standard" that I'm aware of for tumor sequencing data. This "pre-reviewed" data could provide another high quality data set (though only with two samples) for establishing precision.

Response: We thank the reviewer for this great suggestion. We have run Moss in conjunction with Mutect2 on these data, and compared the called variants with the provided *gold list*. Specifically, we have used the whole genome sequencing (WGS) primary tumor and relapse samples as well as the matched normal sample. Since the gold list was derived from the custom targeted capture data, we only focused on the regions covered by these data (with a minimum coverage of at least 100X in each tumor and normal sample).

The candidate list obtained after Running Mutect2 in each tumor sample in isolation generates 14,343 candidates, out of which 1,480 are contained in the gold list (Fig. 4a of the main manuscript). Whereas Mutect2 identifies 1,342 variants from the gold list, Moss identifies 1,396, increasing the recall (Fig. 4b of the main manuscript).

Mutect2 calls 1,679 additional variants, and Moss 3,409 (out of which 1,648 are common to Mutect2). We used the custom targeted capture data to validate these additional variants. Specifically, we say that a SNV is validated if it has a VAF smaller than 0.05 in the normal sample and at least 5 reads with the variant allele in either the primary or relapse sample. From the 3,409 SNVs called by Moss and not contained in the gold list, we found that 1,663 meet the criteria, with a subset of 984 uniquely identified by Moss (Fig. 4c of the main manuscript). From the $1,679 - 1,648 = 31$ SNVs not in the gold list and uniquely identified by Mutect2, only one SNV met the criteria.

We believe these results further demonstrate the ability of Moss to identify low-frequency variants. We have added these results in the revised version of the manuscript. Specifically, we have added the following:

Evaluating Moss on an acute myeloid leukemia dataset with a manually curated list of SNVs.

We test the performance of Moss on an acute myeloid leukemia (AML) dataset. In this dataset, a normal sample, a primary tumor sample and a relapse sample were sequenced with multiple sequencing strategies, including whole genome sequencing (WGS, median coverage of 312×), whole exome sequencing (WXS, median coverage of 433×), and custom targeted capture (median coverage of 1500×). Griffith et al. analyzed the target capture data to produce a manually curated set of high-quality SNVs designated as the gold list. Here, we run Moss in conjunction with Mutect2 on the WGS data. To enable validation, we restrict our attention to candidate SNVs occurring in the genomic regions covered by the targeted capture data (with a minimum coverage of 100× in each of the primary, relapse and normal sample). Running Mutect2 on each WGS tumor sample in isolation yields a total of 14,343 candidates, 1,480 of which occur in the gold list (Fig. 4a). While Mutect2 recalls 1,342 variants from the gold list, Moss recalls 1,396 variants (Fig. 4b). The single SNV from the gold list identified by Mutect2 but not by Moss has low base quality scores of the mutated base, and is subsequently filtered out (Supplementary Fig. 10). Moss additionally identifies 3,409 variants not present in the gold list. To verify these SNVs, we examine their VAFs in the custom targeted capture data. Specifically, we designate a candidate SNV as 'correct' if it has a VAF smaller than 0.05 in the normal sample and at least 5 reads with the variant allele in either the primary or relapse sample, otherwise the SNV is designated as 'incorrect'. 1,663 out of the 3,409 SNVs are correct, with a subset of 984 uniquely identified by Moss (Fig. 4c). Moreover, 30 out of the 31 uniquely identified by Mutect2 (that are not in the gold list) are designated as 'incorrect'. Hence, these findings further confirm Moss' ability to leverage multi-sample data to accurately identify low-frequency SNVs. In addition, we find that Moss decreases the number of SNVs that are unique to one of the (temporal) samples (Supplementary Fig. 11).

Other notes/questions of lesser importance:

Comment 2.3 It would be quite interesting to see what happens if you ran variant callers like strelka/mutect with "relaxed" parameters (rather than default params), then apply Moss. Does that allow for detection of high quality low-VAF variants that would have otherwise been absent from all input VCFs (and thus never made it into Moss)? Evidence that this works (with reasonable precision) would make it an even more powerful tool.

Response: We run Mutect2 and Strelka2 with relaxed parameters to generate the candidate list. However, when analyzing their performance when run without Moss, we use the default settings to identify the called variants. We have made this point clearer in the revised version of the manuscript. Specifically, we have added in the "Overview of the method" the following:

"Moss then extracts the set of candidate loci by taking the union of the positions of all the SNV records in the VCF (variant call format) files (obtained under filtering criteria that are more permissive than the default parameters), as well as the normal alleles inferred by the base caller."

Comment 2.4 The data sets are fairly low coverage by modern standards (though I didn't see coverage info for the CRC samples). Exomes are often sequenced to 200-300x, especially when heterogeneity is being studied. (I definitely understand that samples with 23 biopsies are a unique and rare resource, but tumors with 2-5 samples and higher depth sequencing are plentiful). The core question is: How does deeper coverage data (with potentially more true and false positives at low VAFs) affect the ability of Moss to accurately infer true somatic events? In essence, this question is asking "how low can you go?"

Response: We have downsampled the HCC dataset (original average coverage 75x) to 30x, 20x and 10x, and run Moss in conjunction with Mutect2 sequentially with an increasing number of samples in each of the downsampled datasets. The results are summarized in Supplementary Fig. 8, and show that Moss still recovers a significantly number of variants with coverage as low as 10x. We have also summarized the findings in the main manuscript. Specifically, we have added the following at the end of section "Evaluating Moss in a hepatocellular carcinoma dataset":

"In order to verify Moss' ability to work on datasets with low coverage, we downsample the HCC dataset (original coverage 75x) to 30x, 20x, and 10x, and then rerun Mutect2 in isolation and in conjunction with Moss incrementally on the 23 samples. We find that Moss retains the ability to recover additional variants on low-coverage datasets (Supplementary Fig. 8)."

Comment 2.5 The software is well documented and has a permissive license, which is appreciated. My only note about code is that a docker container would go a long way towards making your software more easily usable in modern sequencing pipelines!

Response: We have included a Code Availability section where we specify the license and the github link containing the code. We have also added it to conda and created a docker image. Installation instructions are available on the github page (<https://github.com/elkebir-group/Moss>).

Comment 2.6 Is this model generalizable enough to cover small indels as well? That extension certainly isn't required for this publication, but would be interesting to see.

Response: The current implementation of Moss does not cover small indels, but we agree that this extension would be beneficial (as also highlighted by Reviewer #1). In particular, such an extension must account for sequencing/mapping artifacts that are specific to indels, for example the presence of homopolymers or low complexity regions in general. We included a comment in the Discussion section:

“There are several directions for future research. First, the concept of using multiple samples to improve variant calling accuracy is broadly applicable beyond single-nucleotide variants. For instance, Zaccaria et al. recently demonstrated that copy number aberrations can be identified with greater accuracy when considering multiple samples from an individual tumor. We expect that the detection of small indels and larger structural variants will benefit from a similar multi-sample analysis as employed by Moss.”

Another interesting and related future direction is to apply the same principles to long-read sequencing data that have elevated error rates compared to short-read sequencing:

“Fourth, it will interesting to adapt Moss to support long-read sequencing data with increased error rates.”

REVIEWERS' COMMENTS

Reviewer #2 (Remarks to the Author):

I'm glad to see that manual review was useful, in allowing you to add additional filtering steps. If I understand the results of manual review in Figure I correctly, even after filtering Moss has a $(32 + 11)/145 = 29\%$ false positive rate, which is quite high.

In the test in the AML data set, I believe that we see the same issue: "1,663 out of the 3,409 SNVs are correct", again suggesting high sensitivity, but also a false positive rate that is very high. That is going to be an issue when used in real world cancer genomics applications.

At the end of the day, I don't want to reject this manuscript, because I believe the tool has value, but would feel better if the manuscript made its limitation more clear. Calling it a high-sensitivity multi-sample caller (instead of high accuracy) would probably be more representative of its true performance.

Without further refinement, I think this tool is going to most useful in a hypothesis-generating context where low-frequency variants will be targeted for additional validation. Accepting the results of this caller without that additional data would probably be a bad idea, and lead to incorrect conclusions downstream. (especially on clinical data, where false negatives are more acceptable than false positives). Doing some edits to the introduction and discussion to avoid overselling this algorithm would probably be warranted.

Point-by-point Response to Reviewers

Nature Communications manuscript NCOMMS-20-22872

We thank the reviewers for their time spent reviewing our manuscript. We believe that this new submitted version addresses all the reviewers' comments. In particular, following Reviewer #2's suggestions, we have updated the abstract, introduction and discussion. A detailed point-by-point response is provided below.

Reviewer #1

Comment 1.1 No comments

Reviewer #2

Comment 2.1 I'm glad to see that manual review was useful, in allowing you to add additional filtering steps. If I understand the results of manual review in Figure I correctly, even after filtering Moss has a $(32 + 11)/145 = 29\%$ false positive rate, which is quite high.

In the test in the AML data set, I believe that we see the same issue: "1,663 out of the 3,409 SNVs are correct", again suggesting high sensitivity, but also a false positive rate that is very high. That is going to be an issue when used in real world cancer genomics applications.

At the end of the day, I don't want to reject this manuscript, because I believe the tool has value, but would feel better if the manuscript made its limitation more clear. Calling it a high-sensitivity multi-sample caller (instead of high accuracy) would probably be more representative of its true performance.

Without further refinement, I think this tool is going to most useful in a hypothesis-generating context where low-frequency variants will be targeted for additional validation. Accepting the results of this caller without that additional data would probably be a bad idea, and lead to incorrect conclusions downstream. (especially on clinical data, where false negatives are more acceptable than false positives). Doing some edits to the introduction and discussion to avoid overselling this algorithm would probably be warranted.

Response: We agree with the reviewer that calling Moss a high-sensitivity multi-sample caller (instead of high accuracy) is more representative of its true performance. As such, we have modified the abstract, introduction, results and discussions to reflect this change. The changes have been highlighted in red such that they can be easily identified.